# Comparative Transcriptomics Reveal Key Sheep (*Ovis aries*) Hypothalamus LncRNAs that Affect Reproduction

**DOI:** 10.3390/ani9040152

**Published:** 2019-04-08

**Authors:** Zhuangbiao Zhang, Jishun Tang, Ran Di, Qiuyue Liu, Xiangyu Wang, Shangquan Gan, Xiaosheng Zhang, Jinlong Zhang, Wenping Hu, Mingxing Chu

**Affiliations:** 1Key Laboratory of Animal Genetics and Breeding and Reproduction of Ministry of Agriculture, Institute of Animal Science, Chinese Academy of Agricultural Sciences, Beijing 100193, China; zhangzhuangbiao18@163.com (Z.Z.); tjs157@163.com (J.T.); dirangirl@163.com (R.D.); liuqiuyue@caas.cn (Q.L.); xiangyu_wiggle@163.com (X.W.); 2Institute of Animal Husbandry and Veterinary Medicine, Anhui Academy of Agricultural Sciences, Hefei 230031, China; 3State Key Laboratory of Sheep Genetic Improvement and Healthy Production, Xinjiang Academy of Agricultural and Reclamation Sciences, Shihezi 832000, China; shangquangan@163.com; 4Tianjin Institute of Animal Sciences, Tianjin 300381, China; zhangxs0221@126.com (X.Z.); jlzhang1010@163.com (J.Z.)

**Keywords:** sheep, hypothalamus, fecundity, lncRNAs

## Abstract

**Simple Summary:**

The hypothalamus has an important role in sheep reproduction. In this study, the key long noncoding RNAs (lncRNAs) associated with sheep fecundity were detected and characterized using the RNA Sequencing technique in sheep hypothalami. The results indicated that several key lncRNAs may affect crucial reproductive processes by differentially influencing the expression of their target genes in polytocous sheep in the follicular phase (PF) vs. monotocous sheep in the follicular phase (MF) and in polytocous sheep in the luteal phase (PL) vs. monotocous sheep in the luteal phase (ML). These results provide an insight into the prolificacy mechanism in sheep without FecB mutation in terms of the hypothalamus.

**Abstract:**

The diverse functions of long noncoding RNAs (lncRNAs), which execute their functions mainly through modulating the activities of their target genes, have been have been widely studied for many years (including a number of studies involving lncRNAs in the ovary and uterus). Herein, for the first time, we detect lncRNAs in sheep hypothalami with FecB++ through RNA Sequencing (RNA-Seq) and identify a number of known and novel lncRNAs, with 622 and 809 found to be differentially expressed in polytocous sheep in the follicular phase (PF) vs. monotocous sheep in the follicular phase (MF) and polytocous sheep in the luteal phase (PL) vs. monotocous sheep in the luteal phase (ML), respectively. Then, Gene Ontology (GO) and Kyoto Encyclopedia of Genes and Genomes (KEGG) analyses were performed based on the predicted target genes. The most highly enriched GO terms (at the molecular function level) included carbonyl reductase (NADPH), 15-hydroxyprostaglandin dehydrogenase (NADP+), and prostaglandin-E2 9-reductase activity in PF vs. MF, and phosphatidylinositol-3,5-bisphosphate binding in PL vs. ML was associated with sheep fecundity. Interestingly, the phenomena of valine, leucine, and isoleucine degradation in PL vs. ML, and valine, leucine, and isoleucine biosynthesis in PF vs. MF, were present. In addition, the interactome of lncRNA and its targets showed that MSTRG.26777 and its *cis-*targets ENSOARG00000013744, ENSOARG00000013700, and ENSOARG00000013777, and MSTRG.105228 and its target WNT7A may participate in the sheep reproductive process at the hypothalamus level. Significantly, MSTRG.95128 and its *cis-*target Forkhead box L1 (FOXG1) were shown to be upregulated in PF vs. MF but downregulated in PL vs. ML. All of these results may be attributed to discoveries of new candidate genes and pathways related to sheep reproduction, and they may provide new views for understanding sheep reproduction without the effects of the *FecB* mutation.

## 1. Introduction

Reproduction involves many organs and expressed genes. Strict regulation of the central nervous system (CNS) and endocrine system, especially related to hormone synthesis and release from the hypothalamus–pituitary–gonadal axis [1], is necessary to achieve reproduction. The gonadotrophin releasing hormone (GnRH) neurons located at the hypothalamus produce GnRH hormones, which can act on the pituitary, triggering the release of follicle-stimulating and luteinizing hormones which both serve as regulators of the ovary by sustaining the functions of the corpus luteum, driving folliculogenesis and synthesizing progesterone (P4) and estrogen (E2) [2,3]. Therefore, GnRH released by the hypothalamus as an initiator of reproduction performs fundamental roles in the modulation of reproduction. Although the importance of GnRH for reproduction has been proven, the mechanisms by which GnRH expression is increased have not been elucidated [4].

Small Tail Han Sheep (STH), an indigenous sheep breed with hyper-fecundity (normally having more than one offspring per birth), has been investigated over many years, especially regarding its reproductive traits. Also, several major fecundity genes, such as the bone morphogenetic protein receptor IB (*BMPR-IB*) gene [5], the bone morphogenetic protein 15 (*BMP15*) gene, and the growth differentiation factor 9 (*GDF9*) gene [6] have been discovered. In addition, some mutations can also influence sheep fecundity significantly. *FecB*, which is a point mutation of *BMPR IB* at A746G, can lead to amino acid mutation from glutamine to arginine [7]. Based on the effects of the *FecB* mutation, which can significantly increase ovulation and litter size [7], STH can be grouped into three genotypes: *FecBB*, with two copies of the *FecB* mutation, *FecB+*, with one copy of the *FecB* mutation, and *FecB++*, with no copies of the *FecB* mutation. In most cases, STH with the *FecB++* genotype have only one offspring per birth. However, in some cases, STH carrying *FecB++* have more than one offspring per birth [8], therefore other potential mechanisms remain to be understood. 

The rapidly-developed technique of advanced RNA Sequencing allows for an in-depth analysis of noncoding RNAs, which have long been regarded as junk but which account for more than 98% of human genome genes [9]. Among those noncoding RNAs, long noncoding RNAs (lncRNAs) refer to a class of RNAs with lengths of more than 200 nt and without the ability to code proteins. LncRNAs can be divided into several types: enhancer RNAs, the heterogeneous class of intergenic transcripts, and the so-called sense or antisense transcripts [10]. As interest toward lncRNAs has increased, the diversity of their functions has been revealed, for example, the role of *Oskar* in oogenesis [11], *Txs* in spermatogenesis [12], *Neat1* in pregnancy [13], and lncRNAs near the *Sxl* promoter in sex determination [14] by *cis*- or *trans*-regulation. Recently, studies have also identified many lncRNAs in sheep and goats. Miao et al. [15] detected several lncRNAs related to STH fecundity; Ren et al. [16] identified lncRNAs associated with skeletal muscle development; and Gao et al. [17,18] primarily identified several lncRNAs that are present in the goat hypothalamus at the puberty phase and then compared the differences between goats and rats in terms of the lncRNA compositions expressed in the hypothalamus during puberty. All of these studies demonstrated that lncRNAs play critical roles in modulating diverse physiological processes, especially the reproduction process. 

The current study explored the lncRNAs that affect STH reproduction in the hypothalamus via RNA-Seq for the first time, providing a foundation from which to investigate the effect of the hypothalamus on the transcriptomic mechanisms responsible for different litter sizes in sheep. 

## 2. Material and Methods

### 2.1. Animal Processing

All of the animals involved in this study were approved by the Science Research Department (in charge of animal welfare issues) of the Institute of Animal Sciences, Chinese Academy of Agricultural Sciences (IAS-CAAS) (Beijing, China) and ethical approval was given by the Animal Ethics Committee of the IAS-CAAS (No. IASCAAS-AE-03). Initially, all of the sheep were fed at a sheep farm of the Tianjin Institute of Animal Sciences, and were all treated similarly with free access to water and feed. In total, 890 Small Tail Han sheep were genotyped in pre-experiment and 142 sheep with the *FecB*++ genotype were identified according to the genotyping results using the TaqMan MGB probe [19]. Then, 12 *FecB++* Han sheep were treated with controlled internal drug releasing (CIDR, progesterone 300 mg) to achieve synchronous estrus. The ovulation rate was also determined by detecting the number of corpus luteum using the laparoscopy procedure in day 6 after CIDR removal. Finally, the sheep were divided into two groups according to their littering records and ovulation rate (Table 1): polytocous sheep (*n* = 6, the litter size and ovulation number was more than one) and monotocous sheep (*n* = 6, the litter size and ovulation number was only one).

### 2.2. Tissues Acquirement and Sequencing

Three polytocous sheep and three monotocous sheep were slaughtered within 45–48 h of CIDR removal (follicular phase), and the remaining six sheep (three polytocous sheep and three monotocous sheep) were slaughtered on day nine after CIDR removal (luteal phase). Finally, all those sheep were divided into two groups according to their littering records and estrous cycle: polytocous sheep in the follicular phase (PF, *n* = 3), polytocous sheep in the luteal phase (PL, *n* = 3), monotocous sheep in the follicular phase (MF, *n* = 3), and monotocous sheep in the luteal phase (ML, *n* = 3).

Hypothalamus tissues were obtained from sheep killed by euthanasia and immediately stored at −80 °C. Then, the stored tissues were used for RNA extraction with TRIzol Reagent (Invitrogen, Carlsbad, CA, USA) according to the manufacturer’s instructions. Examination involving the integrality and quality of the isolated RNA was performed via electrophoresis and the RNA Nano 6000 Assay Kit of the Bioanalyzer 2100 system (Agilent Technologies, CA, USA). The lncRNA library was constructed with 3 μg of high-quality RNA using the NEBNext^®^ Ultra™ RNA Library Prep Kit for Illumina^®^ (#E7530L, NEB, USA) according to the manufacturer’s recommendations. During this process, Ribo-Zero™ GoldKits was used to remove rRNA. In addition, the RNA concentration was assessed, and the RNA library was then sequenced at a concentration of 1 ng/μL RNA using the Illumina platform.

### 2.3. Transcriptome Assembly

The raw reads were generated by sequencing and filtered using the criteria for removing low-quality reads including adapter contamination and reads with *N*% (the percentage of base which cannot be identified by sequencing in a raw read.) of more than 5%. Finally, cleaned reads were obtained. We used the *Ovis aries* reference genome (Oar_v3.1) and the genome annotation file from ENSEMBL; cleaned reads were then mapped to the reference genome using HISAT2 [20]. StringTie [21] was used to assemble the transcripts.

### 2.4. LncRNA Identification and Differential Expression Analysis

Novel IncRNAs with lengths of more than 200 nt and exon numbers greater than two were distinguished with CNCI [22], CPC [23], PFAM [24], and CPAT [25] software after transcriptome assembly. The fragments per kilobase per million mapped reads (FPKM, [26]) values were calculated to represent the expression levels of the lncRNAs. DESeq [27] was then applied to identify the differential expression of the lncRNAs using two comparisons: PF vs. MF and PL vs. ML. In addition, |Log_2_Foldchange |> 0.58, *p* < 0.05 was considered to be a significantly differential expression. 

### 2.5. Target Gene Prediction of lncRNAs and Gene Ontology and Kyoto Encyclopedia of Genes and Genomes Analyses

In order to better understand the functions of differentially expressed lncRNAs in PF vs. MF and PL vs. ML, we carried out target gene predictions. The target genes can be divided into two types, *cis*-targets and *trans*-targets, based on their method of function. *Cis*-targets for lncRNAs execute important regulatory roles on nearby genes that are located less than 50 kb from them. On the other hand, *trans-*targets are lncRNAs that are indispensable for the site of transcription, but they function from remote target genes [9]. Additionally, when the expression quantity correlation coefficient of a lncRNA and its corresponding target mRNA was *p* ≥ 0.9, it was considered to be a potential *trans*-target.

In addition, we performed Gene Ontology (GO) and Kyoto Encyclopedia of Genes and Genomes (KEGG) analyses by using predictable targeted genes of differentially expressed lncRNAs. The hypergeometric test method was applied to assess significantly enriched GO terms and KEGG pathways, and those with *p* < 0.05, after multiple tests and corrections were thought to be significantly enriched.

### 2.6. Construction of Integral LncRNA–mRNA Interaction Networks

To further reveal the potential roles of lncRNAs that are involved in modulating the reproductive process, integral interaction networks containing lncRNAs and their corresponding target genes in PF vs. MF, and PL vs. ML that showed evidence of both *cis*- and *trans*-forms of regulation were built using Cytoscape software [28].

### 2.7. Data Validation

Six lncRNAs—ENSOARG00000025327, ENSOARG00000026466, ENSOARG00000025579, MSTRG.158551, MSTRG.235458, and MSTRG.135103—were selected to validate the accuracy of RNA sequencing via the reverse-transcription quantitative polymerase chain reaction (RT-qPCR). The primers were designed using Primer3 Input (http://bioinfo.ut.ee/primer3-0.4.0/) and synthesized by Tian Yi Biotech. cDNA was used to perform RT-PCR after reverse transcription from total RNA to cDNA. The qPCR reaction conditions were as follows: 95 °C for 15 min, followed by 40 cycles of 95 °C for 10 s and 60 °C for 30 s. The data obtained from qPCR reaction was then calculated using the 2^−△△Ct^ method and processed by SPSS19.0 with a one-way analysis of variance. The results are presented as means ± standard deviation. Furthermore, *p* < 0.05 was regarded as statistically significant.

## 3. Results

To identify differentially expressed lncRNAs (DE lncRNAs) in PF, PL, MF, and ML, an RNA library was constructed. In total, 123,001,527, 117,193,065, 123,812,583, and 122,744,344 raw reads in PF, PL, MF, and ML (on average) were obtained, respectively, from the hypothalamic tissues (Table 2), and the mapping rates were calculated as 92.26%, 92.86%, 92.36%, and 92.55%, on average, respectively, after low-quality reads had been removed and the reads had been mapped to the reference genome in four groups. In addition, regions in the genome with the identified lncRNAs were also predicted (Figure 1A, Appendix A). We can see that many lncRNAs originate from intergenic regions, followed by intron and exon regions. As Figure 1B shows, most of the lncRNAs range from 200 nt to more than 8000 nt, and the majority of lncRNAs have two exons. We also identified novel lncRNAs using CNCI, CPC, PFAM, and CPAT software, and the intersections of these predictions revealed 34,293 novel lncRNAs (Figure 1C, Appendix A). Furthermore, we drew a density plot (Figure 1D), which showed that the pattern of FPKM distribution for the lncRNAs were similar. In addition, the proportions of expression at FPKM ≥ 500 were 1.41% and 10.16% for known and novel lncRNAs, respectively. There were 94.53% and 77.24% FPKM ≤ 50 for known and novel lncRNAs, respectively. From this data, we can see that the expression levels of known lncRNAs (1.41%) at high levels were less than those of novel lncRNAs (10.16%). Regarding DE lncRNAs, we detected 622,809 DE lncRNAs in total, with 406,439 being upregulated and 216,370 being downregulated within the two comparisons (Figure 2, Appendix A). Furthermore, the expression density of DE lncRNAs is also displayed in Figure 3, indicating clearly different expression patterns between PF and MF and between PL and ML. The validation results for the six lncRNAs (Table 3) selected to substantiate the accuracy of sequencing are displayed in Figure 4; the results indicate that there is a similar expression pattern of lncRNAs generated from RNA-Seq and RT-qPCR data.

Subsequently, GO and KEGG analyses were conducted using the identified target genes. As shown in Figure 5A,B and Appendix A, the top three enriched GO terms were carbonyl reductase (NADPH), 15-hydroxyprostaglandin dehydrogenase (NADP+), and prostaglandin-E2 9-reductase activity in PF vs. MF, and the catalytic complex, phosphatidylinositol-3,5-bisphosphate binding, and the growth cone in PL vs. ML. The top 10-enriched KEGG pathways are shown in Figure 5B,D and Appendix A; the most highly enriched pathway was cocaine addiction. Furthermore, many pathways related to metabolism were also enriched, such as valine, leucine, and isoleucine degradation. Regarding PL vs. ML, the most enriched pathway was that of glycosaminoglycan biosynthesis-chondroitin sulfate/dermatan sulfate. Some metabolism pathways were also enriched in PF vs. MF, such as valine, leucine, and isoleucine biosynthesis. 

Increasing evidence and research is demonstrating that lncRNAs function by targeting corresponding genes. For example, NPCCAT1 (a lncRNA) can upregulate *YY1* expression, which can further promote the progression of nasopharyngeal carcinoma [29]. The interaction of the lncRNA Gm2044 with *Utf1* plays an inhibitory role in *Utf1* translation during spermatogenesis [30]. Thus, it is necessary to predict the target genes for lncRNAs and analyze their functions in order to have a complete understanding of the lncRNA functions. Integral networks of lncRNAs and their target genes, containing *cis*- and *trans*-regulatory relationships, were thus built to explore their effects on reproduction (Figure 6). There were 10 lncRNA–gene pairs in PF vs. MF, among which MSTRG.26777 *cis*-regulated four genes. We also constructed 24 lncRNA–gene pairs in PL vs. ML, among which 19 pairs displayed *cis*-regulation and five pairs showed *trans-*regulation.

## 4. Discussion

We initially identified a number of lncRNAs in the hypothalamus from PF, MF, PL, and ML via RNA-seq. We also analyzed the lengths and exon numbers (3448 nt and 2.5 exons, on average, respectively) of the detected lncRNAs. Compared with mRNAs, the lengths of the lncRNAs were smaller [31]; furthermore, the lengths and exon numbers of lncRNAs in sheep hypothalamus tissues were greater than those of the goat hypothalamus tissue (1108 nt and 2.2 exons on average, respectively) [17]. Furthermore, compared with mouse (500 nt) lncRNAs, the lengths of the lncRNAs in the sheep hypothalamus tissues were longer; however, the exon numbers in the sheep hypothalamus tissues were lower than in mouse (3.7, on average) [28]. Therefore, lncRNAs are both tissue- and species-specific [17,32] with diverse phenotypic and functional consequences. 

The top three enriched GO terms in PF vs. MF (at the MF level) were carbonyl reductase (NADPH; ENSOARG00000013744, ENSOARG00000013700, ENSOARG00000013777, Carbonyl reductase 3 (CBR3)), 15-hydroxyprostaglandin dehydrogenase (NADP+) (ENSOARG00000013744, ENSOARG00000013700, ENSOARG00000013777), and prostaglandin-E2 9-reductase (ENSOARG00000013744, ENSOARG00000013700, ENSOARG00000013777) activity. Espey et al. [33] revealed that carbonyl reductase can indirectly modulate the ovulation process in rat ovaries, but the detailed mechanisms of this remain to be explored. Importantly, prostaglandin-E2 9-reductase, a member of the aldo–keto reductase superfamily, possesses wide substrate specificity, including the production of quinones and ketones [34]. The activity of prostaglandin E2-9-ketoreductase and the production of progesterone increased by around six-fold at the time of ovulation [35], which implies that an unclear co-operative relationship may exist. Additionally, progesterone appears to be a trigger for increasing the activity of prostaglandin E2-9-ketoreductase in sheep. Interestingly, progesterone can also increase the activity of 15-hydroxyprostaglandin dehydrogenase [36]. Importantly, a close relationship between NADPH-dependent carbonyl reductase and prostaglandin-E2 9-reductase exists [37]. Along with having similar enriched genes, carbonyl reductase may also be influenced by progesterone, or it may affect its catalytic activity [38] in the hypothalamus. Our results suggest that the activity of carbonyl reductase, 15-hydroxyprostaglandin dehydrogenase, and prostaglandin-E2 9-reductase in the hypothalamus is more intense in PF than that in MF. Given the crucial role of progesterone in GnRH release [39], we hypothesize that carbonyl reductase, 15-hydroxyprostaglandin dehydrogenase, and prostaglandin-E2 9-reductase may play key roles in sheep reproduction through the influence of secreted GnRH. Interestingly, three novel genes—ENSOARG00000013744, ENSOARG00000013700, and ENSOARG00000013777—are *cis*-targets of MSTRG.26777, and they were the top three significantly enriched GO terms (at the MF level), which suggests that MSTRG.26777 has key roles in reproduction. However, this remains to be validated. 

In PL vs. ML, the most greatly enriched GO term of the MF aspect was phosphatidylinositol-3,5-bisphosphate binding (ENSOARG00000015106, Sorting nexin 3 (SNX3), ENSOARG00000011950, SH3 and PX domains 2B (SH3PXD2B), ENSOARG00000018646, ATPase cation transporting 13A2 (ATP13A2), ENSOARG00000012741). Phosphatidylinositol-3,5-bisphosphate binding of genes such as *Sac3* can downregulate the degree to which it responds to insulin in adipocytes [40]. Coincidentally, accumulating evidence [41] suggests that insulin plays a stimulatory role in GnRH release. Phosphatidylinositol-3,5-bisphosphate binding, therefore, may perform important functions in reproduction through its influence on the release of GnRH. SNX3, a member of the sorting nexin (SNX) protein family, is a *trans-*target of MSTRG.39635, and it exerts fundamental roles in protein traffic [42,43]. In the classic action of ovulation modulation, the progesterone receptor and the estrogen receptor, as two critical factors eliciting GnRH release, need to be trafficked to the membrane to act as required [37], and SNX3 may be the one to function in this process. Therefore, we speculate that PL sheep need greater protein transportation activity in the hypothalamus than ML sheep in order to maintain normal luteal functions. ATP13A2, also called transmembrane lysosomal P5-type ATPase, which is *cis*-regulated by MSTRG.130348, is normally located in the lysosome. Matsui et al. [44] proved that cathepsin D activity decreases in cells with ATP13A2-knockdown. Furthermore, prostaglandin F2α and prolactin both function in the rat corpora luteal, with the involvement of cathepsin D [45]. Given the presence of cathepsin D [46], prostaglandin F2α [47], and prolactin [48] in the hypothalamus, an as-yet unknown relationship between the three may exist. All of these differences between PL and ML may affect luteal function and potentially, the transition from the luteal to the follicular phase.

Interestingly, two pathways, named “valine, leucine, and isoleucine biosynthesis” and “valine, leucine and isoleucine degradation”, were enriched in PL vs. ML, and in PF vs. MF, respectively. This demonstrates that some potential roles of valine, leucine, and isoleucine exist in the context of reproduction. Downing et al. [49] demonstrated that an increase of ovulation rate in ewes occurred when a mixture of the branched-chain amino acids leucine, isoleucine, and valine was infused during the late luteal phase of the estrous cycle. However, our results show that the pathways of valine, leucine, and isoleucine biosynthesis, which may occur in the early luteal phase, are found in the hypothalamus. This phenomenon suggests that an increased ovulation rate may be associated with some factors such as amino acids (AAs) biosynthesis in the hypothalamus. Furthermore, arginine, leucine, and glucose can drive the translation and synthesis of proteins, resulting in an increase in the proliferation of ovine trophectoderm cells, mainly through activator components for the mammalian target of rapamycin (mTOR) signaling pathway [50]. Coincidently, this pathway (Wnt family member 7A (WNT7A), ENSOARG00000002034, Eukaryotic translation initiation factor 4E family member 1B (EIF4E1B), Ribosomal protein S6 kinase A2(RPS6KA2), Wnt family member 9A (WNT9A), Frizzled class receptor 2(FZD2), RPTOR independent companion of MTOR complex 2 (RICTOR), NPR3 like, GATOR1 complex subunit (NPRL3), ENSOARG00000011164, Ras related GTP binding D(RRAGD), Mitogen-activated protein kinase associated protein 1 (MAPKAP1), Serum/glucocorticoid regulated kinase 1 (SGK1), Telomere maintenance 2 (TELO2) was also enriched in PL vs. ML. This may be because the mTOR signaling pathway also mediates the modulation of leucine [51], so similar mechanisms may exist in PL vs. ML. WNT7A, a critical WNT family member, is a *cis-*target of MSTRG.105228, and it plays important roles in reproduction. Homozygous Wnt7a/mice can survive, but they display a deficiency in female reproductive duct development; this defect leads to infertile female mice [52]. Recently, studies have proven that WNT7A can promote neural cell proliferation, and even the number of neurons, according to both in vivo [53] and vitro [54] evidence. NPRL3, also known as nitrogen permease regulator-like 3, which is regulated by MSTRG.152806, can greatly affect the neuronal structure and morphology of human, as proven by neural cell lines with NPRL3 knockdown in vitro [55]. Furthermore, the localization of mTOR at the lysosome was also altered in this experiment. It was also found that NPRL3 is necessary for mTOR to modulate AA levels, which may further influence the production process. All of these differences may be responsible for, at least in part, the luteal differences between PF and MF, from the point of view of AAs. 

Additionally, the valine, leucine, and isoleucine degradation pathway (ENSOARG00000004046, Propionyl-CoA carboxylase subunit alpha (PCCA), 3-hydroxy-3-methylglutaryl-CoA lyase (HMGCL), Dihydrolipoamide dehydrogenase (DLD), Alanine--glyoxylate aminotransferase 2 (AGXT2) was enriched in PF vs. MF. AAs, as one of major pathways of function, can be transformed into metabolites such as gamma-aminobutyric acid (GABA) [56], nitric oxide (NO) [57], dopamine [58], and serotonin (5-HT) [59]. The ability of GABA to modulate GnRH secretion has been proven in sheep [60] and mice [61], and its role in driving ovulation has been confirmed [62]. Furthermore, NO, an important metabolite, can affect GnRH release [63]. Dopamine [64,65] and serotonin [66] also influence GnRH, thereby raising the possibility that the degradation of AAs, such as leucine, isoleucine, and valine, in polytocous STH at the follicular phase is greater. In other words, more metabolites are needed for polytocous sheep to maintain their relatively higher fecundity than in motonocous sheep. AGXT2, also known as alanine-glyoxylate aminotransferase 2, which is modulated by MSTRG.81001, can significantly affect the concentration of methylarginine in humans [67]. Importantly, methylarginine can also modulate the activity of neuronal nitric oxide synthase [68]. Considering the key role of nitric oxide in GnRH activation [69], AGXT2 may mediate sheep reproduction by indirectly influencing GnRH release. 

All in all, we speculate that valine, leucine, and isoleucine biosynthesis is fundamental for sheep reproduction in both polytocous or motonocous sheep, and it may be a requirement for maintaining normal luteal function or for conservation of the luteal phase for the degradation of AAs at the follicular phase, according to our results and the experiments discussed above. Also, their activity differs between polytocous and motonocous sheep regarding the degree to which they presumably function in GnRH release, which may be a reason for the difference in litter size.

Considering the way in which lncRNAs function, interactome networks involving lncRNAs and their target genes were predicted to confirm their reproductive roles. Except for the key genes that are regulated by the lncRNAs discussed above, an interesting phenomenon is that MSTRG.95128 and its target gene *FOXG1*, which is enriched in the FoxO signaling pathway, were upregulated in PF vs. MF, but downregulated in PL vs. ML. Regarding *FOXG1*, current studies have mainly focused on its role in neurons. The development of the vertebrate olfactory system, which is necessary for the formation of GnRH neurons in the early development stages [70], requires *FOXG1* expression. *FOXG1*, in the presence of *Emx2*, can play an inhibitory role in gliogenesis and a promotive role on neurogenesis [71], and it can drive the survival of postmitotic neurons [72]. From the KEGG analysis, we know that many metabolites, including GABA, NO, dopamine, and serotonin, produced from AAs, function mainly through signal transductors; therefore, all of the functions of *FOXG1* may facilitate the modulation of neuronal activities, such as those of GnRH neurons. In addition, some researchers have reported that *FOXG1* can influence neural development through the mediation of TGFβ pathways [73]. Considering the involvement of the TGFβ superfamily in GnRH activity, it is speculated that some unexplored relationships between *FOXG1* and GnRH release may exist. Their up- or downregulations may exert different functions on hormone release. 

## 5. Conclusions

To the best of our knowledge, this is the first time that several key lncRNAs in the hypothalamus of *FecB++* STH have been identified in PF vs. MF and PL vs. ML, via RNA-Seq. We also revealed some critical GO terms, interesting pathways, and interactomes between lncRNAs and their target genes through which several lnRNAs and genes were found to influence sheep reproduction. All of these findings may improve the understanding of neuronal transductor processes, especially GnRH release, and we expect this study to provide insight into the mechanisms of sheep hyper-fecundity using the *FecB++* genotype.

## Figures and Tables

**Figure 1 animals-09-00152-f001:**
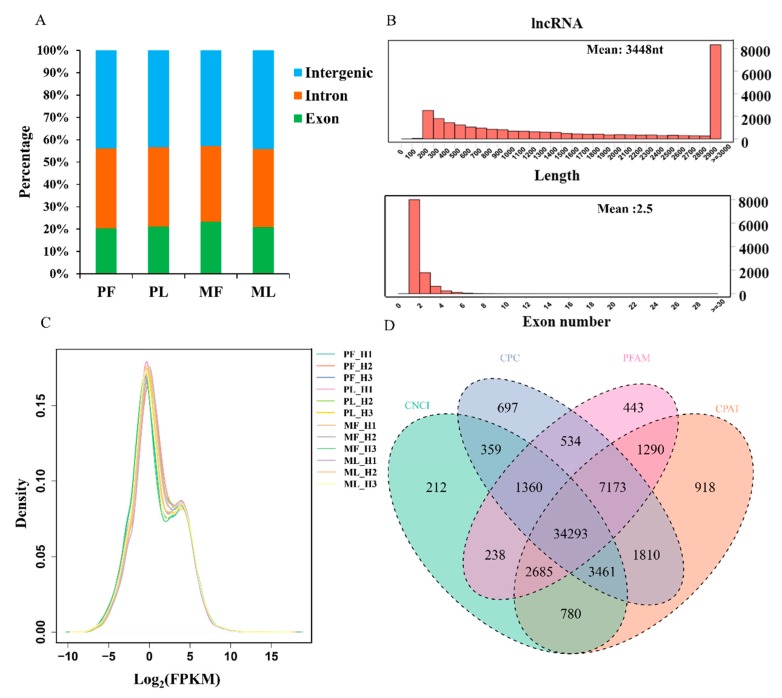
(**A**) The regions of identified long noncoding RNAs (lncRNAs) in polytocous sheep in the follicular phase (PF), polytocous sheep in the luteal phase (PL), monotocous sheep in the follicular phase (MF), and monotocous sheep in the luteal phase (ML) from the genome aspect. (**B**) Distribution of lncRNAs according to length and exon number. (**C**) Distribution of sample expression, which is represented by the fragments per kilobase per million mapped reads (FPKM) value. (**D**) The results of novel lncRNA predictions by using CNCI, CPC, PFAM, and CPAT.

**Figure 2 animals-09-00152-f002:**
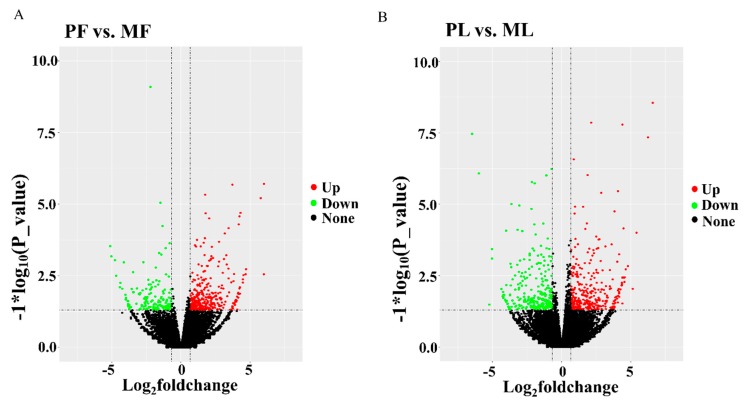
The volcano plots in polytocous sheep in the follicular phase (PF) vs. monotocous sheep in the follicular phase (MF) (**A**) and polytocous sheep in the luteal phase (PL) vs. monotocous sheep in the luteal phase (ML) (**B**), where red and green represent up- or downregulation, respectively.

**Figure 3 animals-09-00152-f003:**
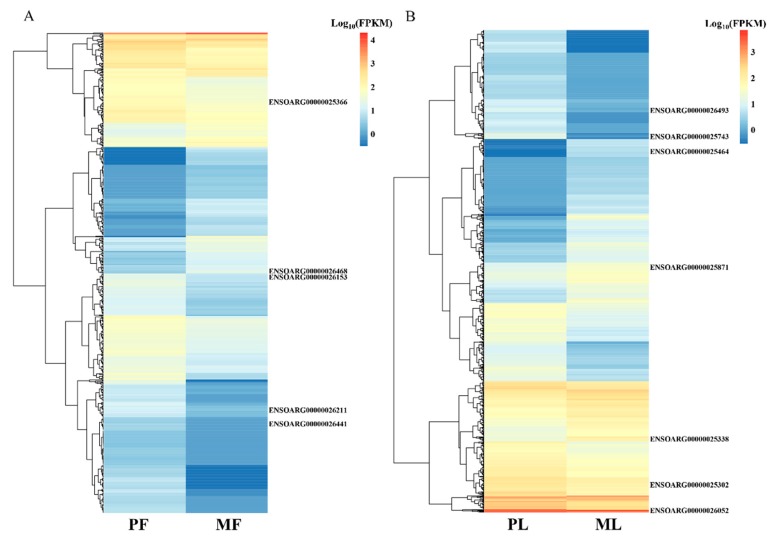
Heat maps showing the expression intensity of 622 and 809 differentially expressed lncRNAs in the follicular phase including polytocous sheep in the follicular phase (PF) and monotocous sheep in the follicular phase (MF) (**A**), and the luteal phase including polytocous sheep in the luteal phase (PL) and monotocous sheep in the luteal phase (ML) (**B**), where all the expressions of lncRNAs (FPKM values) were normalized by log10 (FPKM), and the known lncRNAs were also labeled.

**Figure 4 animals-09-00152-f004:**
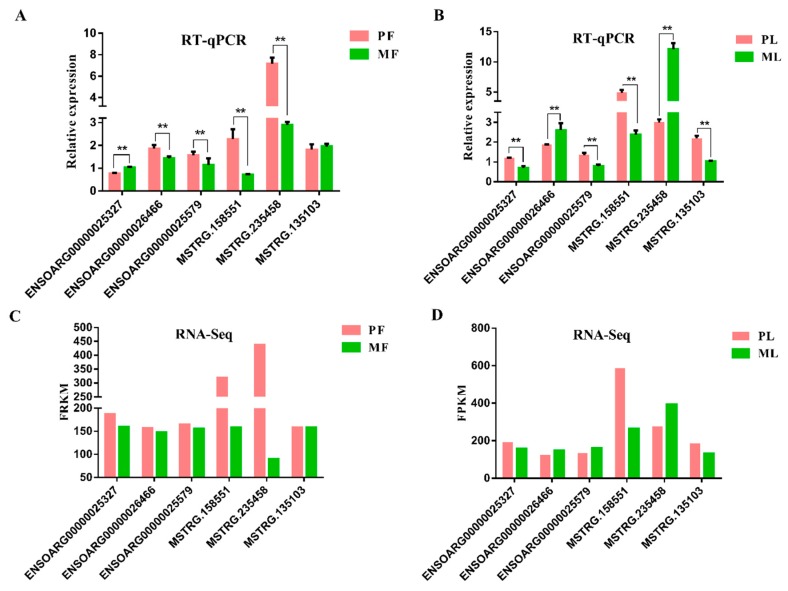
Validation of RNA-Sequencing (RNA-Seq) data using reverse transcription real-time quantitative polymerase chain reaction (RT-qPCR) (**A**,**B**), RT-qPCR results of six selected lncRNAs in polytocous sheep in follicular phase (PF), monotocous sheep in follicular phase (MF), polytocous sheep in the luteal phase (PL) and monotocous sheep in the luteal phase (ML) (**C**,**D**), RNA-Seq results of six selected lncRNAs in PF, MF, PL and ML.

**Figure 5 animals-09-00152-f005:**
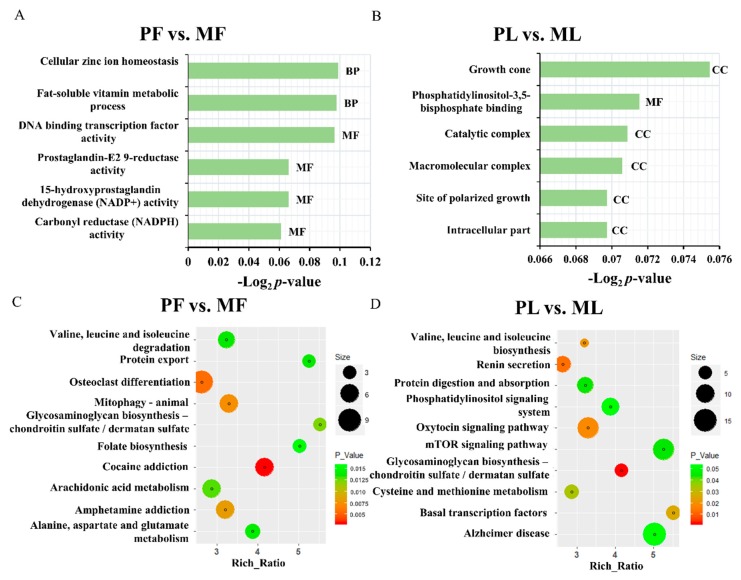
Top enriched Gene Ontology (GO) terms in polytocous sheep in the follicular phase (PF) vs. monotocous sheep in the follicular phase (MF) (**A**) and polytocous sheep in the luteal phase (PL) vs. monotocous sheep in the luteal phase (ML) (**C**). The top 20 enriched Kyoto Encyclopedia of Genes and Genomes (KEGG) pathways in PF vs. MF (**B**) and PL vs. ML (**D**).

**Figure 6 animals-09-00152-f006:**
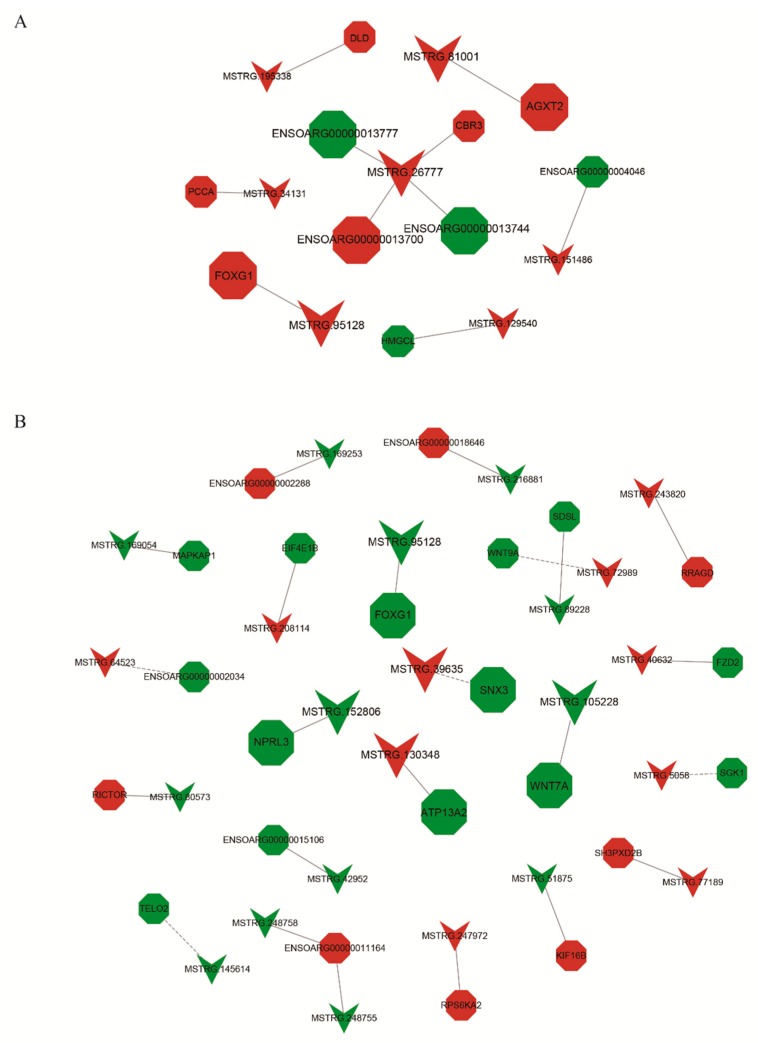
The interaction networks of lncRNAs and their corresponding target genes in polytocous sheep in the follicular phase (PF) vs. monotocous sheep in the follicular phase (MF) (**A**) and polytocous sheep in the luteal phase (PL) vs. monotocous sheep in the luteal phase (ML) (**B**), where the dashed and solid lines represent *trans*- and *cis*-regulation functions, respectively; red and green represent up- and downregulation, respectively; and circles and inverted triangles represent mRNAs and lncRNAs, respectively. Furthermore, bigger circles or inverted triangles indicate that these mRNAs or lncRNAs will be discussed in the Discussion section.

**Table 1 animals-09-00152-t001:** The basic information of ovulation rate and litter size.

Group	Sheep (Earmark)	Litter Size			Ovulation Rate
		First Parity	Second Parity	Third Parity	
Polytocous group	4177	3	3	3	3
	4775	2	3	3	3
	211	3	3	3	3
	265	2	3	3	3
	446	3	3	3	3
	12	3	3	3	3
Monotocous group	4053	1	1	1	1
	4205	1	1	1	1
	3048	1	1	1	1
	4298	1	1	1	1
	3390	1	1	1	1
	4282	1	1	1	1

**Table 2 animals-09-00152-t002:** Summary of the mapping data from the hypothalamic tissues.

Items	Total Reads	Mapped Reads	Mapping Rate	UnMapped Reads	MultiMap Reads	MultiMap Rate
PF_H1	122,545,762	112,451,378	91.76%	10,094,384	5,101,242	4.16%
PF_H2	119,960,438	111,318,118	92.80%	8,642,320	4,914,792	4.10%
PF_H3	126,498,382	116,661,576	92.22%	9,836,806	4,706,940	3.72%
PL_H1	125,264,852	116,254,380	92.81%	9,010,472	5,134,340	4.10%
PL_H2	96,925,068	90,129,243	92.99%	6,795,825	4,194,309	4.33%
PL_H3	129,389,274	120,052,226	92.78%	9,337,048	5,431,242	4.20%
MF_H1	120,588,746	111,485,736	92.45%	9,103,010	4,572,173	3.79%
MF_H2	123,669,806	114,267,242	92.40%	9,402,564	5,236,126	4.23%
MF_H3	127,179,196	117,301,094	92.23%	9,878,102	6,219,896	4.89%
ML_H1	122,378,740	113,525,900	92.77%	8,852,840	4,388,514	3.59%
ML_H2	121,575,158	112,668,594	92.67%	8,906,564	4,770,077	3.92%
ML_H3	124,279,134	114,588,456	92.20%	9,690,678	4,778,914	3.85%

PF, PL, MF, and ML represent polytocous sheep at follicular phase, polytocous sheep at the luteal phase, monotocous sheep at the follicular phase and monotocous sheep at the luteal phase, respectively, and H represents the hypothalamus.

**Table 3 animals-09-00152-t003:** Real-time quantitative polymerase chain reaction primers and sizes of the amplification products of the selected lncRNAs and housekeeping genes.

LncRNA	Primer Sequence	Product Size (bp)
ENSOARG00000025579	CCTCTGAAGCTGCGTGTGTA	241
	TGGGGAGTGATTGAAGTCGG	
ENSOARG00000025327	AGTCCTTGTTCTGCTTGGGG	176
	GCTTGACCATCTGCTCACCT	
ENSOARG00000026466	CCACAGCAAACTCAACGACC	157
	TGCCATCAGAGAAAGAGCCG	
MSTRG.158551	AGGAAAGGCTGATGGTGGTG	238
	CCTCGGGCTTTGTCTCCATT	
MSTRG.235458	TGATGGGAGGTTAGCTGGGA	169
	TACGCACGGTTTGGTTGGTA	
MSTRG.135103	GGAGGTAGAGGGCAAAAGGT	197
	GGGGAAGCAGAAACACAAGG	
β-actin	CCAACCGTGAGAAGATGACC	97
	CCCGAGGCGTACAGGGACAG	

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
