# Peer review of "Comparative Transcriptomics Reveal Key Sheep (Ovis aries) Hypothalamus LncRNAs that Affect Reproduction"

_animals, 2019, doi:10.3390/ani9040152_

Round 1
Reviewer 1 Report
This manuscript is well written, based on sound science and with adequate results output and dense discussion. The subject is important, and the author put a hard work in the experiments. I have several minor comments and revision in the manuscript, but I have one major concern, that can be only a lack of detailed information or it can be a failure in experimental design (in my opinion). The authors used sheep FecB++ (supposed to be with one offspring) and divided between Mono or Polytocous. They inform that use littering records and ovulation rate to do this division, but hey don´t insert additional information regarding this. And this is very important since they further compare the animals in groups according to this division. I have concern in divide sheep according to past records of littering. How past births they consider? This is very irregular, sheep can alternate between one or more offspring during it entire reproductive life. What about the number of lambs (2 or 3) it was evaluated somehow? Please insert full detail regarding the ovulation rate data. Why you decided to use only FecB++ animals? Why no to compare FecB to FecB+ and FecB++ sheep? If this major concern was resolved I think that the manuscript is suitable for publication after minor revision (see attached file)
Author Response
Responds to the reviewer’s comments
Response to Reviewer 1:
This manuscript is well written, based on sound science and with adequate results output and dense discussion. The subject is important, and the author put a hard work in the experiments.
Response: We thank you very much for your positive comments on our manuscript.
1. The authors used sheep FecB++ (supposed to be with one offspring) and divided between Mono or Polytocous. They inform that use littering records and ovulation rate to do this division, but hey don´t insert additional information regarding this. And this is very important since they further compare the animals in groups according to this division. I have concern in divide sheep according to past records of littering. How past births they consider? This is very irregular, sheep can alternate between one or more offspring during it entire reproductive life. What about the number of lambs (2 or 3) it was evaluated somehow? Please insert full detail regarding the ovulation rate data.
Response: Thank you for your comments. Regarding the past record of littering, as you concerned, which may be irregular, so, in order to guarantee the accuracy and reasonability of this study, we recorded the real ovulation rate by detecting the number of corpus luteum using the laparoscopy procedure in day 6 after CIDR removal (only once) and litter size of three parities continuedly to determine the polytocous group (more than two offspring per birth) and monotocous group (only one offspring per birth). (The details of litter size see Table 1 below).
Table 1 The basic information of ovulation rate and litter size
Group | Sheep (Earmark) | Litter size | Ovulation rate | ||
First parity | Second parity | Third parity | |||
Polytocous Group | 4177 | 3 | 3 | 3 | 3 |
4775 | 2 | 3 | 3 | 3 | |
211 | 3 | 3 | 3 | 3 | |
265 | 2 | 3 | 3 | 3 | |
446 | 3 | 3 | 3 | 3 | |
12 | 3 | 3 | 3 | 3 | |
Monotocous Group | 4053 | 1 | 1 | 1 | 1 |
4205 | 1 | 1 | 1 | 1 | |
3048 | 1 | 1 | 1 | 1 | |
4298 | 1 | 1 | 1 | 1 | |
3390 | 1 | 1 | 1 | 1 | |
4282 | 1 | 1 | 1 | 1 |
2. Why you decided to use only FecB++ animals? Why no to compare FecB to FecB+ and FecB++ sheep?
Response: Thank you for your constructive comments. FecB, which is a point mutation of BMPR IB at A746G, can lead to amino acid mutation from glutamine to arginine. Based on the effects of the FecB mutation, which can significantly increase ovulation and litter size, Small Tail Han Sheep (STH) can be grouped into three genotypes: FecBB, with two copies of the FecB mutation, FecB+, with one copy of the FecB mutation, and FecB++, with no copies of the FecB mutation. In most cases, STH with the FecB++ genotype have only one offspring per birth. However, in some cases, STH carrying FecB++ have more than one offspring per birth, and there are almost no publications focus on this mechanism. So, the significance of this study was to investigate the prolificacy mechanism in sheep without FecB mutation from hypothalamus level. Therefore, we selected FecB++ animals in our experiment.
3. You are that luck or you used more sheep and not concluded in this study?
Response: Thanks for your nice suggestion. About this question, here, I want to say that 890 sheep were genotyped in pre-experiment and 142 sheep with FecB++ genotype were selected by the TaqMan probe, and 12 sheep including polytocous sheep at follicular phase (n=3), monotocous sheep at the follicular phase (n=3), polytocous sheep at the luteal phase (n=3) and monotocous sheep at the luteal phase (n=3) were finally determined according to litter size and estrus cycle.
4. N%?
Response: The N% means the percentage of base which cannot be identified by sequencing in raw reads. N% > 5% were regarded low-quality reads and would be removed.
Others revised details have been displayed in the revised article.

Reviewer 2 Report
In this study the authors use RNA-Seq to characterise lncRNA expression in the hypothalamus of small tail han sheep with FecB++ genotypes for variation in litter size.
Generally the paper is interesting and the findings are sound and of novelty particularly to those working on transcription and/or reproductiv traits in sheep.
Please could the authors always include line numbers when submitting manuscripts, it makes reviewing it much easier!
Simple summary - line 1 remove ‘, a critically reproductive tissue,’
Simple summary - second last line remove ‘in-depth’
Abstract - second line remove ‘however’ and start a new sentence here with ‘How lncRNAs are involved in the mechanism….’
Introduction - second paragraph second line insert ‘fecundity’ before ‘genes’
Introduction - second paragraph line 5 remove ‘the prominent one’
Introduction - second paragraph third last line remove ‘where’
Introduction - third paragraph first line, the technique is not that recent, it was first published in 2005, and instead of high throughput sequencing it would be clearer to change this to RNA-Sequencing which is the technique used.
Introduction - last line could be changed to ‘providing a foundation from which to investigate the effect of the hypothalamus on the transcriptomic mechanisms responsible for different litter sizes in sheep’
Animal Processing - Line 6 please give the details or a citation for the Taqman probe used to genotype the sheep.
Transcriptome assembly - first line remove ‘better known as raw reads’ and change ‘after’ to ‘by’ and second line remove ‘out’. Line 3 change ‘cleared’ to ‘cleaned’ and the same in line 4. Line 5 remove ‘continually’.
Data Validation – the primer sequences and qPCR reaction conditions should be included in this section.
Results – first paragraph last line, please could the authors indicate is the RNA-Seq data and RT-qPCR corresponded for all of the lncRNAs measured? As the graph just shows the qPCR results it is difficult to tell and there is no description in the text.
Discussion line 4-5 I don’t see the relevance of the comparison with skin here? The paper referenced is for human rather than sheep. The following comparison with goat is more robust as it is comparing the same tissue and livestock species.
Sheep lncRNA is actually comparable with human if the size of lncRNA for sheep =1108nt and human =1000nt, or at least there is not the same size difference as between human and mouse (500nt) that the authors are suggesting. Similarly 2.5 (sheep) is not significantly lower than 2.9 (human) for the number of exons. The comparison here with mouse is more feasible, the larger mammals appear all to be quite similar.
Discussion – first paragraph second last line change to ‘lncRNAs are both tissue- and species- specific [17,27] with diverse phenotypic and functional consequences.’
Conclusions second last line remove ‘in-depth’ and ‘eliciting’
There is no data availability statement, before publciaiton this dataset should be deposited in the public repositories.
Author Response
Responds to the reviewer’s comments
Response to Reviewer 2:
In this study the authors use RNA-Seq to characterise lncRNA expression in the hypothalamus of small tail han sheep with FecB++ genotypes for variation in litter size. Generally, the paper is interesting and the findings are sound and of novelty particularly to those working on transcription and/or reproductive traits in sheep.
Response: We thank you very much for your positive comments on our manuscript.
1. Please could the authors always include line numbers when submitting manuscripts, it makes reviewing it much easier!
Response: Thanks for your constructive suggestion. The line numbers will be included in next submitting.
2. Simple summary - line 1 remove ‘, a critically reproductive tissue,’
Response: Thanks for your nice suggestion. The ‘a critically reproductive tissue’ has been removed in present article.
3. Simple summary - second last line removes ‘in-depth’
Response: Thanks for your nice suggestion. The ‘in-depth’ has been removed in present article.
4. Abstract - second line remove ‘however’ and start a new sentence here with ‘How lncRNAs are involved in the mechanism….’
Response: Thanks for your nice suggestion. The sentence ‘however’ has been removed, and ‘which execute their functions mainly through modulating their target genes’ has been added.
5. Introduction - second paragraph second line insert ‘fecundity’ before ‘genes’.
Response: Thanks for your nice suggestion. The ‘fecundity’ has been inserted into this article.
6. Introduction - second paragraph line 5 remove ‘the prominent one’
Response: Thanks for your nice suggestion. The ‘prominent one’ has been removed in the article.
7. Introduction - second paragraph third last line remove ‘where’
Response: Thanks for your nice suggestion. The ‘where’ has been removed in the article.
8. Introduction - third paragraph first line, the technique is not that recent, it was first published in 2005, and instead of high throughput sequencing it would be clearer to change this to RNA-Sequencing which is the technique used.
Response: Thanks for your constructive suggestion. The ‘recently-developed’ has been changed to ‘rapidly-developed’. The high throughput sequencing has been changed to RNA-Sequencing in the article.
9. Introduction - last line could be changed to ‘providing a foundation from which to investigate the effect of the hypothalamus on the transcriptomic mechanisms responsible for different litter sizes in sheep’
Response: Thanks for your nice suggestion. The last line has been changed to ‘providing a foundation from which to investigate the effect of the hypothalamus on the transcriptomic mechanisms responsible for different litter sizes in sheep’
10. Animal Processing - Line 6 please give the details or a citation for the TaqMan probe used to genotype the sheep.
Response: Thanks for your constructive suggestion. The details of TaqMan probe has been published previously by other members of our group [1], a citation of TaqMan probe has been added in this article.
[1] Liu, Q.Y., Hu, W.P., He, X.Y., Pan, Z.Y., Guo, X.F., Feng, Tao, Cao, G.L., Huang, D.W., He, J.N., Di, R, Cao, X.H., Wang, X.Y., Chu, M.X. Establishment of high-throughput molecular detection methods for ovine high fecundity major gene FecB and their application. ACTA VETERINARIA ET ZOOTECHNICA SINICA, 2017, 48, 39-51.
11. Transcriptome assembly - first line remove ‘better known as raw reads’ and change ‘after’ to ‘by’ and second line remove ‘out’. Line 3 change ‘cleared’ to ‘cleaned’ and the same in line 4. Line 5 remove ‘continually’.
Response: Thanks for your constructive suggestion. The ‘better known as raw reads’ has been removed and ‘after’ has been changed to ‘by’ respectively in the first line; the ‘cleared’ has been changed to ‘cleaned’ in line 3 and 4; the ‘continually’ has been removed from line 5.
12. Data Validation – the primer sequences and qPCR reaction conditions should be included in this section.
Response: Thanks for your nice suggestion. The primer sequences have been included in present article (see table below), and the qPCR reaction conditions were as followed: 95 °C for 15 min, followed by 40 cycles of 95 °C for 10 s and 60 °C for 30 s, which has been added into ‘Data Validation’.
LncRNA | Primer Sequence | Product size (bp) |
ENSOARG00000025579 | CCTCTGAAGCTGCGTGTGTA | 241 |
TGGGGAGTGATTGAAGTCGG | ||
ENSOARG00000025327 | AGTCCTTGTTCTGCTTGGGG | 176 |
GCTTGACCATCTGCTCACCT | ||
ENSOARG00000026466 | CCACAGCAAACTCAACGACC | 157 |
TGCCATCAGAGAAAGAGCCG | ||
MSTRG.158551 | AGGAAAGGCTGATGGTGGTG | 238 |
CCTCGGGCTTTGTCTCCATT | ||
MSTRG.235458 | TGATGGGAGGTTAGCTGGGA | 169 |
TACGCACGGTTTGGTTGGTA | ||
MSTRG.135103 | GGAGGTAGAGGGCAAAAGGT | 197 |
GGGGAAGCAGAAACACAAGG | ||
β-actin | CCAACCGTGAGAAGATGACC | 97 |
CCCGAGGCGTACAGGGACAG |
13. Results – first paragraph last line, please could the authors indicate is the RNA-Seq data and RT-qPCR corresponded for all of the lncRNAs measured? As the graph just shows the qPCR results it is difficult to tell and there is no description in the text.
Response: Thanks for your nice suggestion. The results of RNA-Seq and RT-qPCR have been added into results part as Figure 4. The results indicated that there is a similar expression pattern of lncRNAs generated from RNA-Seq and RT-qPCR data. (see Figure below).
14. Discussion line 4-5 I don’t see the relevance of the comparison with skin here? The paper referenced is for human rather than sheep. The following comparison with goat is more robust as it is comparing the same tissue and livestock species.
Response: Thanks for your nice suggestion. The comparison between hypothalamus and skin has been removed.
15. Sheep lncRNA is actually comparable with human if the size of lncRNA for sheep =1108nt and human =1000nt, or at least there is not the same size difference as between human and mouse (500nt) that the authors are suggesting. Similarly 2.5 (sheep) is not significantly lower than 2.9 (human) for the number of exons. The comparison here with mouse is more feasible, the larger mammals appear all to be quite similar.
Response: Thanks for your nice suggestion. For better suitable for the topic content, the comparison between human and sheep has been removed.
16. Discussion – first paragraph second last line change to ‘lncRNAs are both tissue- and species- specific [17,27] with diverse phenotypic and functional consequences.’
Response: Thanks for your nice suggestion. The first paragraph second last line has been changed to ‘lncRNAs are both tissue- and species- specific [17,27] with diverse phenotypic and functional consequences.’
17. Conclusions second last line remove ‘in-depth’ and ‘eliciting’
Response: Thanks for your nice suggestion. The ‘in-depth’ and ‘eliciting’ have been removed from conclusions second last line.
18. There is no data availability statement, before publciaiton this dataset should be deposited in the public repositories.
Response: Thanks for your nice suggestion. All the RNA-sequencing data used in this study has been deposited in the Sequence Read Archive (SRA) public databases (https://submit.ncbi.nlm.nih.gov/) under BioProject (PRJNA529384).

Round 2
Reviewer 1 Report
I had one major concern regarding the manuscript and the author successfully clarify the methodology and now I am positive that the results presented are according to the defined groups and the sheep used are for sure within the groups. Additionally, the authors correct all my minor suggestions and answer all my doubts regarding the study.
As minor revision request, parts of the text from the reviewer’s report should be inserted in the manuscript, since I believe that is important information to readers.
This information should be included in the methodology:
“I want to say that 890 sheep were genotyped in pre-experiment and 142 sheep with FecB++ genotype were selected by the TaqMan probe....
The author should explain in the manuscript the abbreviation N% at first time used.
Finally, I was checking my initial review report and there is a paragraph at page 11 marked. Please dismiss this, I will suggest a change of location, but I think it´s fine where it is.
The revised version of the manuscript including a new table and more information regarding the study design satisfactorily answer my concerns and I consider the manuscript suitable for publication after minor revision listed above.
Author Response
Response to Reviewer 1:
As minor revision request, parts of the text from the reviewer’s report should be inserted in the manuscript, since I believe that is important information to readers.
1. This information should be included in the methodology:
“I want to say that 890 sheep were genotyped in pre-experiment and 142 sheep with FecB++ genotype were selected by the TaqMan probe....
Response: Thank you for your comments. The information mentioned above has been included in the methodology
2. The author should explain in the manuscript the abbreviation N% at first time used.
Response: Thank you for your comments. Regarding N% (the percentage of base which cannot be identified by sequencing in raw reads.), the explanation of N% has been added into this article.
Thanks again for your constructive comments!